# Chinese Consumers’ Trust in Food Safety Surveillance Sampling of Commonly Consumed Foods

**DOI:** 10.3390/foods11131971

**Published:** 2022-07-02

**Authors:** Xiaocheng Wang, Yanbo Xu, Miaomiao Liang, Jian Gao, Jing Wang, Si Chen, Jingmin Cheng

**Affiliations:** 1School of Management, Shanxi Medical University, Taiyuan 030001, China; buyi21shiji@163.com (X.W.); xuyanbo19980312@163.com (Y.X.); 18335754068@163.com (M.L.); gaojian@b.sxmu.edu.cn (J.G.); dawnwjing@163.com (J.W.); 2School of Public Health, Shanxi Medical University, Taiyuan 030001, China; 3China National Center for Food Safety Risk Assessment, Risk Communication Division, Beijing 100020, China

**Keywords:** supervision sampling inspection, competence trust, care trust, perception of food safety, attitude, commonly consumed foods, China

## Abstract

In China, food has become safer over the past five years, especially commonly consumed foods. Food supervision sampling has played an important role in improving food safety. However, consumer acceptance of the results of food safety supervision have not kept pace. Communicating actual food safety risks to consumers and improving the public trust in food safety supervision sampling inspection has become an important issue. This study focused on food safety surveillance sampling of commonly consumed foods. In total, 4408 adult consumers were surveyed between August and October 2021. Structural equation modeling was performed for data analysis. This study found significant differences along gender lines and across different cities and levels of education with respect to evaluating competence trust and care trust on food supervision sampling inspection. This study identified the public’s competence trust, care trust, and perception of food safety as factors that significantly affect one’s attitude toward supervision sampling inspection. Care trust showed a more pronounced effect on trust enhancement than competence trust. The present study also provides some practical measures for food safety supervisors to improve public trust in the national food inspection. Specifically, the sampling process should be open and transparent.

## 1. Introduction

With the development of the social economy, the standard of living in China has been greatly improved, but food safety problems still frequently appear [1]. People are, however, paying more attention to food safety in China [2], as these food safety problems not only cause physical harm to consumers but also cause psychological panic. The degree of concern about food safety has gradually become an important key to measuring people’s quality of life in China [3]. Consumers of commonly consumed foods are particularly concerned. To this end, the states have developed and introduced a series of food standards to ensure consumer safety [4,5,6]. By 2020, a food safety supervision system based on risk analysis and supply chain management has been established, and major regional and systemic food safety risks have been controlled. There are two types of food safety: objective food safety and subjective food safety, the latter is also known as the perception of food safety (PFS) [7]. Objective food safety refers to a concept based on the assessment of the risk of consuming a certain food by scientists and food experts [8]. Perception of food safety is a person’s perception of the potential risk associated with food safety questions [9], or consumer concern about whether a particular food product can be consumed without harmful effects [10]. At present, the overall situation of food safety in China is generally improving. From 2016 to 2020, the evaluation of food safety supervision sampling inspection showed that the overall pass rate was higher than 96%, particularly, for meat products, processed grain products, edible oils/grease products, dairy products, and egg products. In recent years, the State Administration of Market Supervision and Administration has undertaken a series of risk communication measures to improve food safety. However, consumer’s perception of food safety risk based on subjective psychological factors often deviates from the actual risk level. public confidence in food safety has not improved significantly. Translating food safety risks to consumers and increasing public trust in the national food inspection can be challenging. Finding a way to improve the public trust toward food safety supervision sampling inspection [11] has become a core issue in food safety risk communication.

### 1.1. Literature Review

Since the reform and opening up (1978), food safety management has received increasing attention in China. In particular, since the establishment of the State Administration for Market Regulation in 2018, the scope and content of food safety supervision have been improved and intensified to ensure food quality. The academic understanding of food safety management has increased [12], as various disciplines have become more integrated, the number of people and institutions providing authoritative research has increased [13,14], and food safety management has become an influential academic field. However, due to the impact of major food safety incidents in recent years, the most urgent task for China’s food safety management is to enhance public trust in food safety.

Trust is a complex, multidimensional concept that includes both rational components (derived from experience) and irrational components (based on instinct and emotion) [15,16]. Trust also occurs at two distinct levels: the interpersonal and the institutional [17,18]. Interpersonal trust is negotiated between individuals, for example, between a consumer and a retailer, whereas institutional trust is placed in one or more social systems or institutions (e.g., the Sampling and Monitoring Department of Food Safety, State Administration for Market Regulation, China). Institutional trust is a standard predictor of trust in key institutions in organizations [19] such as the government or legal organizations [20,21]. Institutional trust theory suggests that people’s trust in an institution affects their perceptions of that institution. Research on consumers has extended institutional trust theory to exploring the effects of institutional trust on perceived expertise [22], perceived risk [23], product trust [22], and interpersonal trust [24]. Both interpersonal and institutional trust are important for understanding where and how trust can be (re)developed and maintained in the context of food safety regulation and compliance.

The two-dimensional model of trust is currently widely used in the field of food safety and includes competence trust and care trust [25]. Competence trust refers to trust based on knowledge and performance, which is judged mainly on the past behavior of the trusted object and the possible behavior in the future expectation, reflecting the cognitive component of trust. Care trust refers to trust based on motivation and relationships, which is generally judged based on the closeness of the public’s connection to the trust recipient and inferences about the trust recipient’s intentions and motivations, reflecting the emotional component of trust.

Trust plays a crucial role in risk communication and management, and public trust is also a topic that cannot be ignored in the supervision sampling inspection of food products. Some scholars have proposed that factors affecting consumers’ confidence in food safety include their trust in participants in the food chain [26,27,28,29,30,31] and regulatory authorities [32,33], memories of food safety events, media reports [33,34,35,36], perceptions of the safety of different types of products [28,37,38], and consumers’ demographic characteristics [27,39,40,41] and values [25].

Official government food safety supervision departments are the main bodies that carry out food sampling inspections. Owing to the complexity of the food production system, consumers are not able to accurately judge the safety of food during the food consumption process, leading them to rely on other participants in the food chain to provide them with safe food, as well as on the government and society’s regulatory systems [42]. When consumers believe that the results of food sampling inspections can be trusted and that when food safety problems occur, the authorities will take appropriate action to prevent them from endangering public health, such as issuance of recalls [43], these beliefs have the potential to directly influence the level of consumer optimism about food safety. When carrying out food safety surveillance and sampling inspections, steps such as the development of sampling plans, the development of food safety standards, and the allocation and implementation of work can all reflect the actual competence of the authorities concerned, which translates into competence trust in the trust model [44,45,46,47]. Therefore, the quality of food safety supervision sampling inspection work may affect the public’s attitude toward its results. See appendix file 1 for the relevant concepts for this study.

The demographic characteristics of consumers vary in terms of gender, age, occupation, education status, income level, and socioeconomic status, all of which affect the level of consumer trust in food safety regulation to varying degrees. The more sensitive consumers are to food regulatory information, the more helpful it is to prompt the government to make objective and comprehensive disclosures of health food regulatory information, and the more consumers trust the information [48]. In addition, the overall performance of social trust may vary across societal periods and may have an impact on the perception of government food safety regulations [49]. Therefore, when monitoring the public’s attitude toward food safety supervision sampling inspection results, considering the impact of social trust facilitates accurate measurement of public trust and effective intervention.

The purpose of this study is two-fold:(1)To analyze the differences in public trust regarding food safety supervision sampling inspection of commonly consumed foods in China across different demographic groups;(2)To explore the core focus of enhancing trust in national food inspection, improve the public’s attitude toward the qualified rate of food safety supervision sampling inspection, and put forward improvement measures for food safety supervision sampling inspection.

### 1.2. Hypothesis

#### 1.2.1. Effect of Institutional Trust on Attitude

Institutional trust is comprised of competence trust and care trust. The safety and reliability of food are important reflections of the government’s ability to monitor and manage risk in the market. In the relationship of government trust, the subject is the citizen, and the object is the government. Hetherington [50] incorporated government competence into the connotation of government trust, and the level of public trust in the government to have the knowledge and skills required for its management is a performance-based indicator of trust in the government. Levi [51] posited that the goodwill of the government is an important component of government trust, representing the extent to which the government cares for people’s livelihoods and interests, as measured by the motivations and goals of government management behavior. To a large extent, consumers believe that the government is responsible for ensuring food safety [52] and should take responsibility for disclosing food safety information and communicating risk. National food safety monitoring and inspection departments should regularly publish relevant information and undertake risk monitoring. Consumers’ trust in the work of food safety supervision sampling inspection departments directly affects their attitude toward the results released to the public by these authorities. Yang and Holzer [53] suggested that the public’s approval of government work reflects the public’s trust in government. It is generally believed that the higher the public’s trust in government, the higher the level of satisfaction with the government’s work, and the more credible the information released to the government. Based on the above discussion, the following hypotheses were formulated:

**Hypothesis 1** **(H1).**
*The public’s competence trust in safety supervision sampling inspection has a significant positive effect on the attitude toward the public announcement of the qualified rate of safety supervision sampling inspection of commonly consumed foods.*


**Hypothesis 2** **(H2).**
*The public’s care trust in safety supervision sampling inspection has a significant positive effect on the attitude toward the qualified rate of safety supervision sampling inspection of commonly consumed foods.*


#### 1.2.2. Effect of Institutional Trust on the Perception of Food Safety

Institutional trust is essential for relationships, certifications, and organizational assurance, and can be facilitated by increasing consumer confidence in normative and expected outcomes [54]. Consumers’ perception of food safety is defined as “the consumer’s perceived judgment of the level of food safety under specific circumstances” [55]. Scholars have generally theorized a strong correlation between trust and risk perception [56]. For example, Hu [57] found that populations with a high level of trust in companies using gene technology generally had lower risk perception. When choosing food products, consumers may have specific concerns about safety, hygiene, cleanliness, and the presence of chemical residues [58]. Consumers who purchase food often interact directly with food retailers and indirectly with the food regulators responsible for managing food hazards [59]. De Jonge [11] argued that consumers trust the organizations that form the food supply chain (e.g., producers, manufacturers, and retailers of food) and food regulators (e.g., governments, legislatures, and consumer associations). Previous research has shown that trust affects the perceived food safety of various food products [33]. Researchers have confirmed that trust in government associations affects consumers’ perception of food safety [60]. For example, Feng [61] found that trust in the State Food and Drug Administration had a significant effect on risk perception. Another study showed that the more ineffective the public perceived government regulation of additive safety to be, the higher their level of risk perception of additives and the greater the likelihood of refusal to purchase [62]. This suggests that the effectiveness of government supervision of food safety significantly affects the level of consumers’ perceptions of food safety. Therefore, the following hypotheses were proposed:

**Hypothesis 3** **(H3).**
*The public’s competence trust in supervision sampling inspection has a significant negative effect on the perception of food safety.*


**Hypothesis 4** **(H4).**
*The public’s care trust in supervision sampling inspection has a significant negative effect on the perception of food safety.*


#### 1.2.3. Effect of Perception of Food Safety on Attitude

The relationship between risk perception and attitude has been studied. Consumers’ food safety risk perception is based on a subjective perception of existing risks. The reality of such risks may not necessarily exist but will have a great impact on consumers’ attitudes. Lobb [63] suggested that attitudes towards the product are negatively affected by risk perception based on the SPARTA model. Choi [64] found that consumers’ perception of risk negatively affected their attitude toward street food. However, Dang [65] argued that risk perception had a positive effect on attitude toward traceable foods. The more risk perceived, the more likely consumers could express a positive attitude toward traceable foods. Per common sense, risk perception has a negative impact on attitude regarding common foods [63,64]. We conclude that the lower the public perception of overall food safety risk, the more they agree with the results of the high pass rate recorded by the regulatory agencies. This led us to the following hypothesis:

**Hypothesis 5** **(H5).**
*The perception of food safety has a significant negative effect on the public’s attitude toward the qualified rate of safety supervision sampling inspection of commonly consumed foods.*


#### 1.2.4. Effect of Generalized Trust on Attitude and Perception of Food Safety

Generalized trust is a kind of trust based on similar values and norms that undergird social trust, also known as social trust, i.e., trust in strangers or many people in society. Compared to individualized trust, building social trust is more time-consuming, but costs less and may bring greater social efficiency [62]. The empirical results of one study show that social trust had a significant positive impact on the well-being of the population [66]. Kunitoki [67] found that increasing social trust in the HPV vaccine in Japan led to renewed confidence in the vaccine and a reduction in preventable deaths and complications. Liu [68] explored the impact of social trust on parents’ risk perceptions and vaccination intentions in China, where social trust was negatively associated with perceived risk but positively associated with perceived benefits. Therefore, the following hypotheses were proposed.

**Hypothesis 6** **(H6).**
*The public’s generalized trust has a significant positive effect on the public’s attitude toward the qualified rate of safety supervision sampling inspection of commonly consumed foods.*


**Hypothesis 7** **(H7).**
*The public’s generalized trust has a significant negative effect on the perception of food safety.*


## 2. Materials and Methods

### 2.1. Participants

The survey was conducted online using the Jishuyun Technology network (https://www.databnu.com/ accessed on 1 August 2021), which can provide a professional online questionnaire survey, and the online questionnaire was administered to adults via a WeChat applet. Quota sampling was used in this research, with a total of 4408 consumers to be surveyed in 31 provinces in China. First, the 31 provinces were divided into 7 regions. Second, in every region, the sample was aligned with the composition ratio of the 2020 national census data from the National Bureau of Statistics in terms of gender, and age. Eligible consumers can log into the WeChat applet and voluntarily choose to participate in the survey until the required sample size is reached. Data were collected between August and October 2021.

Under the principle of anonymity and voluntary, the respondents were invited to answer the questionnaire according to their real feelings and situations. The measurement information was only used for scientific research. 

### 2.2. Questionnaires

The online questionnaire (written in Chinese) comprised three sections, for a total of 30 questions. (see Appendix A). Section 1 was related to the socio-demographic information of the participants, including their gender, age, education level, annual per-capita household income, place of residence, and whether they were mainly responsible for food purchasing/cooking in their household. Section 2, the main scale in this study, included trust in supervision sampling inspection (competence trust and care trust) [25], generalized Trust [69,70], perception of food safety, and attitude toward the highly qualified rate of supervision sampling inspection of commonly consumed foods [71]. In Section 3, the strategies that should be taken by the supervision and sampling management to improve the trust in the qualified rate of food safety supervision sampling inspections were measured.

All items were measured on a 5-point Likert scale, with 1 indicating ‘strongly disagree’ and 5 indicating ‘strongly agree.’ A pre-survey was conducted after the questionnaire was compiled to evaluate its feasibility, and to improve the effectiveness of online questionnaire survey. The questionnaire takes approximately five to ten minutes to complete. The questionnaires were reviewed after collection, and polygraph questions were developed (i.e., the same point of view was asked from both positive and negative perspectives to determine the reliability of participants’ responses). Questionnaires with too short a response time (under 3 min) or too consistent in response options (>90% agreement) were excluded. A total of 4408 questionnaires were collected (20% return rate online), of which 4082 were valid, with an effective rate of 92.6%. Although the response rate of the online surveys was lower than that of pen-and-paper and face-to-face surveys, it was consistent with rates reported in the other studies [72]. The sample covered a wide geographical area and was relatively representative, providing a good indication of the true feelings of Chinese consumers about food safety supervision sampling inspection.

### 2.3. Statistical Analysis

First, the consumers’ trust in commonly consumed food safety supervision sampling inspection was described as mean and SD, and differences under different demographic characteristics were analyzed to find the effect of individual characteristics on trust in supervision sampling inspection. If the assumptions concerning the homogeneity of variance and normal distribution were met, the *t*-test and one-way ANOVA test were used for analysis.

Second, Confirmatory Factor Analysis (CFA) was conducted to determine the reliability and validity of the measurement model, and Pearson correlation was used to calculate the correlation coefficient between latent variables.

Third, the theoretical model was analyzed using Structural Equation Modeling (SEM) to assess the overall fit of the model and represent the strength of the effect across different variables using standardized regression coefficients (*β*) and *p*-values.

Finally, we calculated the mean score of each strategy that contributes to trust improvement and ranked them.

Statistical analysis was carried out using SPSS 23.0 and AMOS 17.0 software packages. When *p* < 0.05, differences were considered statistically significant.

## 3. Results

Initially, we found that the differences in competence and care trust between consumers with different demographic characteristics were statistically significant. The public’s care and competence trust were important factors influencing their attitude toward the results of sampling and inspection according to the Structural Equation Model. Ultimately, we discuss the foci that could significantly increase public trust in the supervision sampling inspection of commonly consumed foods.

### 3.1. Socio-Demographic Characteristics

Participants were 50.5% female and 49.5% male consumers. In terms of age, 26.7% of participants were under 30 years old, 20.4% were 30–39 years old, 22.9% were 40–49 years old, 14.6% were 50–59 years old, and 15.4% of respondents were over 60 years old. Most participants had a low level of education, with 38.7% having a technical secondary school or senior high school degree and 22.6% having a junior high school degree or lower. In terms of household income, most of the participants came from middle-income families, with 29.1% of participants earning RMB 20,000–29,999 per year and 20.4% earning RMB 30,000–39,999 per year. Moreover, 49.3% of the participants lived in urban areas, and 50.7% lived in rural areas. Further, 81.4% of the participants were responsible for food shopping or cooking daily (Table 1).

### 3.2. Differences in Trust in Supervision Sampling Inspection by Participant Characteristics

The differences in competence trust and care trust between consumers with different demographic characteristics were statistically significant (*p* < 0.05). As shown in Table 2, the results of the associations between consumer characteristics and competence trust indicate that people who were female, aged 30–49, with a college degree or above, living in a rural area, and being the member of the household primarily responsible for daily food shopping or cooking had the highest score for competence trust in the work of food safety supervision sampling inspection (all *p* < 0.05), but earning RMB 30,000–39,999/year had the lowest score (*p* < 0.05). In terms of the motivational trust, we found no significant variation across consumers who were and were not primarily responsible for daily food shopping or cooking. Other results were consistent with competence trust, except for individuals aged 30–39 years.

### 3.3. Measurement Model

Descriptive statistics of all measurement items in the model are shown in Table 3. Internal consistency reliability was examined using Cronbach’s alpha and composite reliability. Composite reliability relies on actual loadings to calculate factor scores and is a better indicator of internal consistency reliability [73]. As shown in Table 3, the composite reliability values for each sub-construct in the model were above the recommended threshold of 0.7 [74], except for Generalized Trust, thus supporting the reliability of the measure.

The factor loading and Average Variance Extracted (AVE) of all items should be higher than the recommended value of 0.50 [75] (Table 3). In the current study, the AVE was 0.462–0.781, which indicated a good convergent validity. Discriminant validity was examined at both the item and construct levels. As shown in Table 4, at the construct level, the square roots of the AVE were higher than those of the respective correlation coefficients [76]. Thus, discriminant validity was supported.

### 3.4. Structural Equation Modeling (SEM)

The outcome of the SEM presented well-fitted data (χ^2^/*df* = 2.664, GFI = 0.994, AGFI = 0.990, NFI = 0.993, IFI = 0.996, CFI = 0.996). The RMSEA value obtained was 0.020, which is less than the recommended value of 0.08 [77]. Other indices such as CFI, GFI, IF, and NFI all reached standard values of approximately 0.9 and higher [78]. Therefore, the data in this study fit the model appropriately, as evidenced by the good fit of the indicators.

Table 5 shows the ultimate decision of the proposed hypothesis of the model. The *t*-value for the path of H1 (2.587), H2 (12.678), H3 (−5.612), H4 (−6.451), H5 (−5.341), and H7 (−4.146) was higher than the standard value. Therefore, the study findings indicate the existence of statistically significant positive relationships between competence trust (*β* = 0.129, *p* < 0.05), care trust (*β* = 0.736, *p* < 0.01), perception of food safety (*β* = −0.151, *p* < 0.01), and the public’s attitude toward a high qualified rate of supervision sampling inspection. Other significant relationships were observed between competence (*β* = −0.295, *p* < 0.01), care (*β* = −0.431, *p* < 0.01), generalized trust (*β* = −0.146, *p* < 0.01), and perception of food safety. Thus, the outcomes corroborate hypotheses 1–5, and 7. In contrast, generalized trust did not show a significant relationship with the public’s attitude (*p* > 0.05) (see Figure 1). Therefore, hypothesis 6 was rejected.

The public’s care trust and competence trust were important factors influencing their attitude toward the results of sampling and inspection. In terms of the total effect, care trust had a greater weight than competence trust and was the core focus of trust enhancement, as shown in Table 6.

### 3.5. Improvement Measures

At the same time, we studied consumers’ opinions on measures to improve the supervision sampling inspection process. As shown in Figure 2, the foci that could significantly increase their trust in the supervision sampling inspection of commonly consumed foods were as follows: “The sampling process should be open and transparent (IM2: Mean score = 3.63),” “The rigorous regulation should be implemented (IM8: Mean score = 3.58),” and “The most stringent standards should be established (IM7: Mean score = 3.58).”

## 4. Discussion

The two-dimensional models of trust that are currently more widely used in the food safety field are competence trust and care trust. Competence trust refers to trust based on the performance of competence in terms of knowledge, skills, and behavior. Care trust refers to trust based on motivation and relationships and reflects the public’s relationship with the person they trust and their assumptions about the intentions and motivations of the person they trust. We found significant differences by gender, place of residence, and level of education when evaluating trust in food supervision sampling inspection. Women and people living in rural areas showed more focus on food safety than men or city dwellers. The higher the consumer’s level of education, the more they trusted regulators. Consumers’ perceived trust was better when more knowledge and good practices were applied.

The current results indicate that institutional trust has a significant positive effect on the attitude toward the public announcement of the qualified rate of safety supervision sampling inspection of commonly consumed foods. This result is consistent with the finding of Costa-Font M [79], which showed that consumers’ trust in public regulatory authorities is an important factor affecting the public’s attitude toward GM food and reduces their worry. The results also indicate that care trust has greater weight on changing the attitude towards supervision sampling inspection than competence trust. This suggests that food safety supervision and sampling organizations will be more effective in fostering trust in qualified rates by focusing on care trust than on competence trust, which is consistent with the findings of previous studies [80,81]. In a study on topics such as food additives, Chen [80] found that addressing the public’s level of apprehension or feelings of helplessness should increase the public’s care trust in the government rather than competence trust. Supervision sampling inspection should focus on emotion and rationality when communicating with the public, prioritizing emotional responses, demonstrating similar values and similar core concerns, and responding positively to audience concerns [81]. In the meantime, supervision sampling inspection departments should convey the attitude that they are fully considering the public’s interest, enhancing care trust, reducing confrontational interpretations, and thus accumulating core evidence to convince people with reason.

The survey also explored the public trust in supervision sampling inspection. Among the ten efforts to enhance trust, the public most strongly endorsed “The sampling process should be open and transparent”, “The most stringent standards should be established”, and “The rigorous regulation should be implemented”. The third lowest ranking for “The sampling and testing results should be open and transparent” reflects, to some extent, the openness and transparency of the process more than the openness and transparency of the results in terms of trust in food safety supervision sampling inspection.

## 5. Implications

### 5.1. Theoretical Implications

This research has contributed to the extant literature in numerous ways. First, this study takes consumers’ trust in the supervision and sampling work by food regulatory authorities as an independent variable to analyze the impact of consumers’ attitudes toward the qualified rate of supervision and sampling inspection. This study focuses on the work of supervision and sampling, which is an important link in the food safety supervision chain, and evaluates that work from a consumer perspective. This field has not been studied yet, thus, this study fills a gap in the literature. Second, this study focuses on commonly consumed foods. As these foods account for the largest proportion of daily nutritional consumption, they are also likely the foods that consumers care about most. The impact is also greatest if food safety incidents occur. At the same time, it also provides ideas for research on other types of food (e.g., functional or organic food). Third, we construct a trust-to-attitude model to analyze the mediating effect of consumers’ perception of food safety on their trust in food safety regulatory authorities and attitude toward the pass rate. The model also considers the influence of social trust. Our research contributes to the literature by clarifying the relationships among institutional trust, perception of food safety, and attitude toward food safety surveillance sampling of commonly consumed foods.

### 5.2. Managerial Implications

The results of the study suggest that adequate national supervision sampling work may improve consumer trust in food risk screening, as well as promoting public approval of the qualified rate of supervision and sampling inspection. Here are some measures proposed for food safety regulators. In the production of food safety supervision sampling inspection videos, the General Administration can consider the results of perception surveys and prioritize the presentation of views and beliefs shared by the public, such as demonstrating the ability to detect potential food safety hazards and the openness and transparency of the supervision sampling inspection process to promote public empathy and enhance the acceptance and trust of the message. When formulating food sampling and inspection plans, the difference between “sampling and inspection priorities” and “public concerns” can be narrowed appropriately by incorporating commonly consumed food categories that are of high public concern in questionnaires and public opinion surveys to actively respond to audience concerns and enhance public trust in the government’s sampling inspection. It is recommended that the disclosure of information on food items of key concern to the public should be enhanced to meet the information needs of the public in depth. The focus on food information disclosure should be considered a top priority to meet the information needs of the public. When the sampling test results are released, expert interpretations can be combined to clarify professional information that the public should know and want to know but do not know in an easy-to-understand way [82].

China can learn from the more advanced work of other countries. For example, the European Food Safety Authority (EFSA), Europe’s leading food risk management agency, has produced a detailed summary of the different approaches used in its work alongside a guidebook. Its work has also gained public attention, safeguarding the quality of EFSA’s food safety science, and enabling a transparent and trustworthy relationship based on open and effective dialogue. The EFSA classifies audiences and suggests content strategies for communicating with different audiences, as well as providing on the technical and professional skills needed [18]. Not only does this increase the efficiency of communication with the audiences but also increases audience trust in EFSA-related work [18,83].

### 5.3. Policy-Making Implications

Food safety supervision is a systematic project, which transforms the supervision of government departments into a cooperative supervision mechanism dominated by government departments and involving the participation of relevant social forces. To improve the effectiveness of food safety supervision, relevant departments should focus more on consumer concerns when formulating policies. Policymakers should realize that the entire food system has to work toward fulfilling consumers’ needs [84]. The principle of taking consumers as the center is reflected in the sampling inspection, re-inspection, and processing activities organized by the food market supervision and administration department. China’s food safety supervision system of laws and regulations should also be further improved, with strict implementation of food safety laws and regulations. We should strengthen oversight of food safety across the board, establishing the strictest standards, enforcing stringent regulation, imposing the severest penalties, and insisting on the most serious accountability. Our study also found that these strategies could significantly increase public trust in the supervision sampling inspection of commonly consumed foods.

## 6. Limitations and Future Research Directions

This study has several limitations and associated recommendations for further research. First, quota sampling via a WeChat applet may lead to sample selection bias, resulting in biased results that do not represent the entire population. Compared with the structure of the Chinese population, the sample population is similar in age and gender distribution, but there are differences in terms of education status and place of residence (See Appendix A). This may be related to the use of an online survey, as consumers with a low level education may not be comfortable using mobile phones. In the future, a multistage, random cluster process should be used to acquire the sample to be surveyed in each province. Second, cultivating consumer trust is a continuous and dynamic process. Cross-sectional studies can only reflect the situation at a particular period. Longitudinal studies should be designed for further research. Third, perception of food safety depends not only on trust, but also on other variables, such as the different sources of information, the credibility of the information sources, the content of the information, and the quantity and quality of the information. In future studies, we will pay particular attention to these variables to provide more explanations for residents’ perceptions of food safety. Fourth, trust in supervision sampling inspection is crucial to improve the perception of food safety, but it is also worth determining whether trust can improve consumers’ food purchasing behavior. Future research could consider mediating variables such as intention, perceived benefits, or attitude when examining the trust–behavior gap. Additional variables and multivariate statistical analyses could extend the current framework. This provides a new direction for future research. Fifth, owing to the impact of the COVID-19 pandemic, this survey was conducted online, and participants may be more likely to choose not to respond online than when face-to-face.

## 7. Conclusions

To improve public trust in food safety, this present study focused on food safety surveillance and sampling. Results showed significant gender differences, as well as differences according to the city the participants came from or their level of education when evaluating competence trust and care trust on food supervision sampling inspection. This study showed consumers’ competence trust, care trust, and perception of food safety to be the factors that significantly affect the public’s attitude toward the high qualified rate of supervision sampling inspection. Care trust was the core focus of trust enhancement rather than competence trust. Measures for enhancing public trust in national food inspection can be developed based on this research.

## Figures and Tables

**Figure 1 foods-11-01971-f001:**
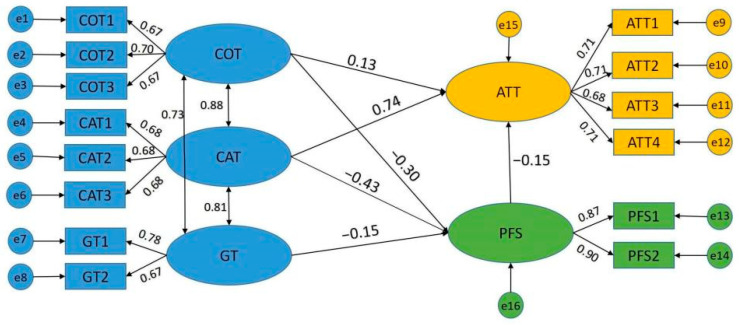
Structural Equation Model. COT: Competence trust; CAT: Care trust; GT: Generalized trust; PFS: Perception of food safety; ATT: The public’s attitude toward the high qualified rate of safety supervision sampling inspection of commonly consumed foods.

**Figure 2 foods-11-01971-f002:**
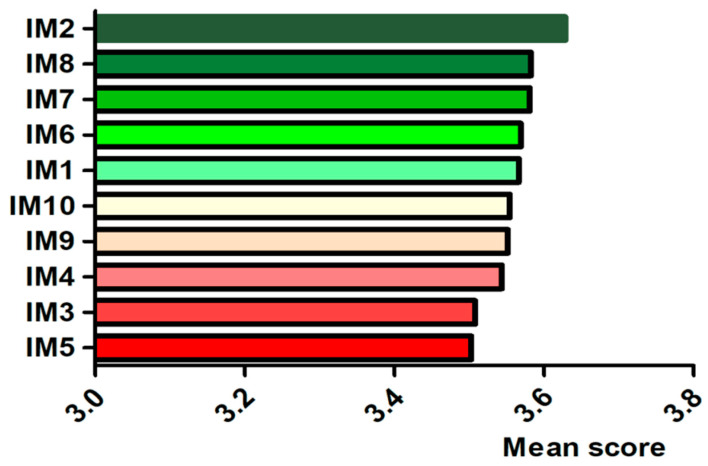
Improvement measures for food safety supervision sampling inspection. Data are presented as mean. IM1: The sampling scheme should be scientific and reasonable; IM2: The sampling process should be open and transparent; IM3: The sampling and analysis techniques should be accurate; IM4: The sampling and testing results should be open and transparent; IM5: The sampling and testing results should be interpreted in detail to respond to public concerns; IM6: Substandard products should be effectively traced and recalled; IM7: The most stringent standards should be established; IM8: Rigorous regulation should be implemented; IM9: The most severe penalties should be imposed; IM10: The most serious accountability should be upheld.

**Table 1 foods-11-01971-t001:** Descriptive statistics of consumers’ demographic characteristics.

Variables	Categories	Frequency (n)	Percent (%)
Gender	Male	2021	49.5
	Female	2061	50.5
Age (years)	<30	1090	26.7
	30–39	831	20.4
	40–49	936	22.9
	50–59	598	14.6
	≥60	627	15.4
Education status	Junior high school degree or below	922	22.6
	Senior high school degree	1580	38.7
	College degree	774	19.0
	Bachelor’s degree or above	806	19.7
Per-capita annual household income from last year (RMB)	<10,000	160	3.9
	10,000–19,999	813	19.9
	20,000–29,999	1188	29.1
	30,000–39,999	831	20.4
	40,000–59,999	515	12.6
	≥60,000	575	14.1
Place of residence	Urban	2012	49.3
	Rural	2070	50.7
Responsible for food shopping or cooking daily	Yes	3322	81.4
	No	760	18.6

**Table 2 foods-11-01971-t002:** Associations between the characteristics of consumers and competence trust and care trust.

Variables	Categories	*n*	Competence Trust	*p*-Value	Care Trust	*p*-Value
Gender	Male	*2021*	3.64 (0.87)	0.013 †	3.73 (0.87)	0.018 †
	Female	*2061*	3.71 (0.84)		3.79 (0.82)	
Age (years)	<30	*1090*	3.60 (0.90) ^a^	0.000 ‡	3.72 (0.88) ^a^	0.000 ‡
	30–39	*831*	3.85 (0.70) ^b^		3.97 (0.69) ^b^	
	40–49	*936*	3.77 (0.80) ^b^		3.86 (0.79) ^c^	
	50–59	*598*	3.50 (0.92) ^c^		3.57 (0.93) ^d^	
	≥60	*627*	3.58 (0.92) ^a^		3.62 (0.91) ^d^	
Education status	Junior high school degree or below	*922*	3.53 (0.94) ^a^	0.000 ‡	3.56 (0.91) ^a^	0.000 ‡
	Senior high school degree	*1580*	3.54 (0.95) ^a^		3.63 (0.95) ^a^	
	College degree	*774*	3.88 (0.64) ^b^		3.99 (0.61) ^b^	
	Bachelor’s degree or above	*806*	3.89 (0.64) ^b^		4.04 (0.59) ^b^	
Per-capita annual household income of last year (RMB)	<10,000	*160*	3.76 (0.68) ^a^	0.000 ‡	3.84 (0.66) ^a^	0.000 ‡
	10,000–19,999	*813*	3.76 (0.77) ^a^		3.87 (0.75) ^a^	
	20,000–29,999	*1188*	3.68 (0.86) ^a^		3.74 (0.84) ^a^	
	30,000–39,999	*831*	3.51 (0.96) ^b^		3.60 (0.96) ^b^	
	40,000–59,999	*515*	3.72 (0.81) ^a^		3.85 (0.80) ^a^	
	≥60,000	*575*	3.69 (0.87) ^a^		3.79 (0.87) ^a^	
Place of residence	Urban	*2012*	3.54 (0.92)	0.000 †	3.65 (0.93)	0.000 †
	Rural	*2070*	3.80 (0.77)		3.87 (0.74)	
Responsible for food shopping or cooking daily	Yes	*3322*	3.69 (0.86)	0.002 †	3.77 (0.85)	0.437 †
	No	*760*	3.59 (0.85)		3.74 (0.82)	

Legend (†) *t*-test; (‡) One-way ANOVA test; ^a–d^ Different letters indicate significant differences (*p* < 0.05) according to the Student–Newman–Keuls test.

**Table 3 foods-11-01971-t003:** Cronbach Alpha, Composite Reliability, AVE.

Constructs	Mean (std Deviation)	Item Loading	AVE	CR	Alpha ɑ
**Competence trust (COT)**			**0.463**	**0.721**	**0.720**
COT1	3.61 (1.038)	0.674			
COT2	3.66 (1.105)	0.696			
COT3	3.75 (1.062)	0.670			
**Care trust (CAT)**			**0.462**	**0.721**	**0.720**
CAT1	3.76 (1.066)	0.680			
CAT2	3.74 (1.053)	0.681			
CAT3	3.79 (1.049)	0.679			
**Generalized trust (GT)**			**0.520**	**0.685**	**0.680**
GT1	4.10 (0.892)	0.777			
GT2	3.90 (0.997)	0.667			
**Perception of food safety (PFS)**			**0.781**	**0.877**	**0.877**
PS1	2.22 (1.183)	0.870			
PS2	2.17 (1.191)	0.897			
**Attitude (ATT)**			**0.495**	**0.797**	**0.797**
ATT1	3.63 (1.041)	0.706			
ATT2	3.67 (1.113)	0.715			
ATT3	3.65 (1.098)	0.685			
ATT4	3.68 (1.111)	0.709			

Note: Average Variance Extracted (AVE).

**Table 4 foods-11-01971-t004:** Correlation of Latent Variables and Square Roots of AVE.

	Competence Trust	Care Trust	Generalized Trust	Perception of Food Safety	Attitude
**Competence trust**	**0.680**				
**Care trust**	0.636 **	**0.680**			
**Generalized trust**	0.510 **	0.570 **	**0.721**		
**Perception of food safety**	0.622 **	0.643 **	0.549 **	**0.884**	
**Attitude**	0.680 **	0.635 **	0.593 **	0.703 **	**0.704**

Note: Bold indicates the square root of AVE. ** Correlation is significant at the 0.01 level (2-tailed).

**Table 5 foods-11-01971-t005:** Structural Equation Model and Hypothesis Testing Result.

Hypotheses	Beta	STD Beta	S.E.	*t*-Values	*p*-Values	Significance (*p* < 0.05)
H1: COT → ATT	0.134	0.129	0.052	2.587 **	0.010	Supported
H2: CAT → ATT	0.762	0.736	0.060	12.678 ***	0.000	Supported
H3: COT → PFS	−0.427	−0.295	0.076	−5.612 ***	0.000	Supported
H4: CAT → PFS	−0.624	−0.431	0.097	−6.451***	0.000	Supported
H5: PFS → ATT	−0.108	−0.151	0.020	−5.341 ***	0.000	Supported
H6: GT → ATT	0.023	0.022	0.036	0.631	0.528	Not Supported
H7: GT → PFS	−0.217	−0.146	0.052	−4.146 ***	0.000	Supported

Note: ** Significant at 5% level, *** Significant at 1% level, STD = Standard.

**Table 6 foods-11-01971-t006:** Direct Effect, Indirect effect, and Total effect (Public’s attitude toward the high qualified rate of safety supervision sampling inspection of commonly consumed foods).

Path	Direct Effect	Indirect Effect	Total Effect
COT → ATT	0.129	0.045	0.174
CAT → ATT	0.736	0.065	0.801
GT → ATT	-	0.022	0.022
PFS → ATT	−0.151	-	−0.151

## Data Availability

The data that support the findings of this study are available from the corresponding authors upon reasonable request.

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
