# Peer review of "Chinese Consumers’ Trust in Food Safety Surveillance Sampling of Commonly Consumed Foods"

_foods, 2022, doi:10.3390/foods11131971_

Round 1

Reviewer 1 Report

General observation:

-       The topic is interesting, but the paper needs major revisions.

Abstract

-       Replace over “18 years of age” with “adults”. Do the same in the text.

-       Add the period of the survey.

-       Keywords: add more keywords, for example, food safety, China, etc.

Introduction

-       You need to add information to explain the recent evolution of food safety practices in China, recent history, the governance of food safety in China, the recent scandals, etc. This will help the reader to understand the importance of your research and your topic.

-      Please merge 1.1 definitions and 1.2 Literature review within the introduction.

-      In your Literature review, where is the research that analyzes China's food safety?

-       The need for the research (i.e. identified gap) was not clearly explained or justified in the introduction. Please address how this paper is original. Based on my previous comments, this can enhance your argument about the importance of studying food safety in China.

Materials and Methods

-       Delete the sub-titles.

-       Give more information about the sampling method.

-       How did you distribute the survey: via social media? Emails?...

-       Include an appendix with the English version of the questionnaire at the end of the article. As a result, you might shorten the questionnaire-related paragraphs. There is no need to include detailed information about the questions in the text, only basic information.

-       Add information about the questionnaire validation? Pre-test?....explain.

Results

-       Add a paragraph with a clear summary of the study findings at the beginning of section 3. Results and before 3.1.

Discussions

-       Compare your results to previous studies.

-       Limitation and future directions: I believe that this sub-section is weak, limitations should be expanded, and it should clearly highlight the shortfalls and future recommendations for future studies.

Conclusions

-       Please add the managerial and theoretical contribution of the paper.

-       Please clarify how the study findings contribute to and extend the literature.

-       This paper lack contribution to policymakers; please develop a section to tackle this issue.

Reviewer 2 Report

Thank you Dear Author for the opportunity to read the text. The text is well written. I just have a comment: please separate the LIMITATIONS section.

Reviewer 3 Report

This is a good quality submission. The research problem is relevant and important. The applied method is correct. The results are presented clearly.

When you describe the socio-demographic characteristics of the sample, please compare them with the general population to see which characteristics of your sample are similar to the structure of the Chinese population, and which are different. What could be the effect of these differences on your results?

What could be the impact of cultural differences on competence trust and care trust?

lines 358-359 - You write about positive relationships, but the Beta is negative. Please clarify.

365 - hypothesis 6

408 - What do you mean by "qualified rates"?

462-463 - Please provide some reference for your statement.

Incomplete references" 5, 14, 17, 18, 20, 52, 74, 76

Round 2

Reviewer 1 Report

Thanks for taking into consideration most of my comments. Here are some minor revisions/suggestions:

-       Add the reference to the new first paragraph that you added in the introduction.

-       In the methodology, add that you adopted the snowball sampling method. Your sample is not representative of the population, as you can see in table S1. So you have to add that in the limitation: the sample bias.
